# Fuzzy Broad Learning System Combined with Feature-Engineering-Based Fault Diagnosis for Bearings

**Jianmin Zhou** [1,2,*] **, Xiaotong Yang** [1,2] **, Lulu Liu** [1,2] **, Yunqing Wang** [1,2] **, Junjie Wang** [1,2] **and Guanghao Hou** [1,2]

1    Key Laboratory of Conveyance and Equipment, East China Jiaotong University, Ministry of Education, Nanchang 330013, China
2    State Key Laboratory of Rail Transit Infrastructure Performance Monitoring and Guarantee, Nanchang 330013, China
*    Correspondence: 1981@ecjtu.edu.cn; Tel.: +86-137-5568-5348

**Abstract:** Bearings are essential components of rotating machinery used in mechanical systems, and fault diagnosis of bearings is of great significance to the operation and maintenance of mechanical equipment. Deep learning is a popular method for bearing fault diagnosis, which can effectively extract the in-depth information of fault signals, thus achieving high fault diagnosis accuracy. However, due to the complex deep structure of deep learning, most deep learning methods require more time and resources for bearing fault diagnosis. This paper proposes a bearing fault diagnosis method combining feature engineering and fuzzy broad learning. First, time domain, frequency domain, and time-frequency domain features are extracted from the bearing signals. Then the stability and robustness indexes of these features are evaluated to complete the feature engineering. The features obtained by feature engineering are used as the input of the fault diagnosis model, and three sets of experimental data validate the model. The experimental results show that the proposed method can achieve the bearing fault diagnosis accuracy of 96.43% on the experimental bench data, 100% on the Case Western Reserve University dataset, and 100% on the centrifugal pump bearing fault dataset, with a time of approximately 0.28 s. The results show that this method has the advantages of accuracy, rapidity, and stability of bearing fault diagnosis.

**Keywords:** bearing fault diagnosis; fuzzy broad learning system; feature engineering; bearing; fuzzy rules

## 1. Introduction

Bearings are one of the most frequently used components operating various large machinery. Nevertheless, in modern industrial working conditions, bearings have faced numerous failure triggers due to the working conditions, which can lead to many safety problems, causing certain losses and impacts on social safety and economic benefits. Therefore, the timely and effective detection of the degree and type of failure of train bearings is of great significance [1].

The general steps for fault diagnosis of bearings consist of three steps: Acquiring fault signals, extracting fault features, and performing fault diagnosis. Numerous scholars have shown through research studies [2–7] that it is feasible to detect bearing faults using motor current frequency, stray flux signals, thermal imaging images, and acoustic emission signals. However, the most practical and proven method is the vibration signal of the bearing, which is easy to capture. The bearing fault diagnosis based on the vibration signal can also identify the bearing fault by focusing on the intrinsic frequency with tiny vibration, thus achieving higher diagnostic accuracy.

Presently, there are many feature extraction methods for vibration signals of bearings, and feature extraction of vibration signals in the time domain, frequency domain, and time–frequency domain is still the primary extraction method [8]. However, effectively

achieving information fusion is a significant difficulty [9]. Zhang Chi [10] et al. calculated the P-values of different features and fault types to reflect the importance of features by using the two-sample Kolmogorov–Smirnov test. Mario Peña [11] et al. constructed a comprehensive validity assessment framework using ANOVA and performed a correlation on the bearing evaluation framework for bearing fault diagnosis. Li [12] et al. enriched the feature space by integrating a feature extractor with a joint attention module to maintain the distinction between different classes of features. Wang [13] et al. proposed an end-to-end feature selection and diagnosis method to achieve the primary feature selection for prediction models by selecting features with large Mean Impact Values (MIV).

Among the data-driven methods based on vibration signals, the three main bearing fault diagnosis methods are vibration signal-based signal decomposition, shallow machine learning, and deep learning. Considering that, in practical applications, to carry out a fast and accurate diagnosis of fault types, many methods often cannot be both in both aspects, the Broad Learning System (BLS) was initially proposed as an alternative to deep learning networks, which are often computationally intensive and time-consuming to train. BLS is built as a planar network that simplifies the network structure by transferring the original input as "mapped features" to feature nodes and extending the structure in width at the "enhanced nodes". Therefore, BLS can improve the model's classification ability while considering the computational speed. There is also a growing number of applications of the BLS and an increase in the number of improved broad learning systems for different objectives. Yu [14] et al. proposed a progressive integration framework called PEKB to integrate multiple KBLs for better accuracy, which can improve the system's noise immunity and generalization capability. Feng [15] et al. proposed a new deep domain knowledge-based broad learning framework to solve homogeneous multitasking problems. Pu [16] et al. combined semi-supervised learning with broad learning to improve the model diagnosis rate by training unlabeled data. Feng [17] et al. fused Takagi-Sugeno fuzzy system with BLS to form the fuzzy broad learning system (FBLS), which reduces the structural complexity of the model and runs better.

However, either integrating the model, deepening the depth of the model, or adding regularization terms will increase the workload required for network computation. Compared with these improved algorithms, FBLS has the following advantages [18]: (1) The fuzzy subsystem included in FBLS makes it better able to handle various fuzzy logic problems, thus extracting a wider range of features for subsequent enhanced mapping and computation of regression results; (2) the fuzzy subsystem uses K-means to determine the number of fuzzy rules in each system, and due to the characteristics of K-means, each fuzzy subsystem produces different affiliation centers for the training data, thus producing different results, which can extract more information from the input data; (3) the parameters to be calculated in FBLS are the weights between the augmented layer and the output layer and the fuzzy initialization coefficients of the polynomials in the successor part of the subsystem, which can be analytically computed, so FBLS retains the fast computational characteristics of the traditional BLS.

This paper proposes a rolling bearing fault diagnosis method that combines feature engineering and a fuzzy broad learning system, which can quickly detect bearing faults and help improve the robustness and accuracy of the bearing fault diagnosis model. This paper extracts the features of the time domain, frequency domain, and time–frequency domain features as input features, where the time–frequency domain feature extraction method is wavelet energy entropy feature extraction. The extracted features are used to calculate the correlation and robustness between the features to eliminate all the poorly performing features in the two evaluation indexes to obtain the better-performing feature sets. The filtered features are fed into the fuzzy broad learning system for bearing fault diagnosis experiments. The experimental data used were a dataset collected by the testbed, a set of bearing datasets from a centrifugal pump, and a bearing dataset from Case Western Reserve University, respectively. Experiments on the model can demonstrate the superiority of the proposed method in terms of training and testing time and fault diagnosis accuracy.

The main structure of this paper is as follows: Section 2 explores the details of the theory. Section 3 describes the specific process of the approach, and the results, as well as the discussion, are stated in Section 4. Section 5 concludes the paper.

## 2. The Basic Theory of the Model

### 2.1. Feature Engineering

The vibration signal of bearings in operation has the characteristics of non-smoothness and non-linearity. It is difficult to accurately identify the fault type directly by fault diagnosis on the original signal, so it is necessary to perform signal extraction first to extract the critical information in the signal that can characterize the fault before subsequent analysis. Signal analysis based on the time domain, frequency domain, and signal analyses in the time and frequency domain is three crucial ways of extracting vibration signals. The time domain and frequency domain features extracted in this paper are the maximum value, minimum value, average value, peak-to-peak value, rectified average value, valid value, peak value, variance, standard deviation, skewness, kurtosis, wave-form factor, peak factor, impulse factor, margin factor, the center of gravity frequency, mean square frequency, frequency variance, root mean square frequency, and frequency standard deviation of the signal, respectively.

The signal time-frequency domain feature extraction method is to perform a three-layer wavelet packet decomposition [19] of the vibration signal using db5 wavelet packets, and its decomposition structure tree is shown in Figure 1. The main process is to obtain $M = (M = 2m)$ sub-signals of length $L/M$ for a signal of sample length $L$, after m-layer wavelet packet decomposition. Each sub-signal corresponds to a frequency interval, and the $i$th sub-signal corresponds to the frequency interval $\left[ f_{s(i-1)}/M, f_{si}/M \right]$, where $f_s$ is the signal sampling frequency, and $i = 1, 2, \cdots, M$.

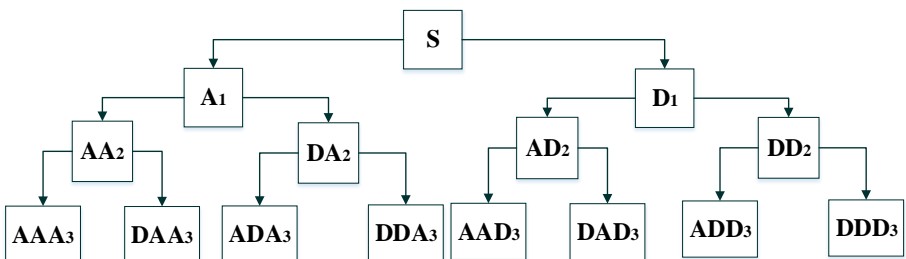

**Figure 1.** Wavelet packet signal decomposition structure diagram.

After using the wavelet packet decomposition method, the signal contains $M$ different frequency bands in the $0 \sim f_s$ interval. Reconstructing the sub-signal coefficients again, the energy of each sub-signal can be calculated as:

$$E_i = \int\limits_{-\infty}^{+\infty} h_i^2(t)dt, (i = 1, 2, \cdots, M) \tag{1}$$

where $h_i(t)$ is the reconstruction coefficient of the $i$th sub-signal after wavelet packet decomposition. With $M$ sub-signals, the energy vectors $[E_1, E_2, \cdots, E_M]$ of different frequency bands after wavelet packet decomposition can be obtained, and the energy values are normalized to obtain the energy ratio as:

$$\eta_i = E_i/E \tag{2}$$

where $E = \sum\limits_{i=1}^{M} E_i$ is the total energy and has $E = \sum\limits_{i=1}^{M} \eta_i = 1$. The energy entropy $P$ can be calculated from Equation (3) as:

$$P = \sum_{i=1}^{M}[-\eta_i \ln(\eta_i)] \tag{3}$$

By extracting wavelet packet energy entropy features from the original vibration signal data, 8 IMF components are obtained. The 8-wavelet packet energy entropy features are numbered 1 to 8, the 20 extracted time and frequency domain features are numbered 9 to 28, and the correlation and robustness of all features are calculated as follows.

$$Corr(F, T) = \frac{\left| K\sum\limits_{k} f_T(k)T_k - \sum\limits_{k} f_T(k)T_k \sum\limits_{k} T_k \right|}{\sqrt{\left[ K\sum\limits_{k} f_T(k)^2 - \left(\sum\limits_{k} f_T(k)\right)^2 \right]\left[ K\sum\limits_{k} (T_k)^2 - \left(\sum\limits_{k} T_k\right)^2 \right]}} \tag{4}$$

$$Rob(F) = \frac{1}{K}\sum_{k}\exp\left(-\left|\frac{f_R(k)}{f(k)}\right|\right) \tag{5}$$

where the order of the feature signals is defined as $F = [f_1, f_2, \cdots, f_K]$. The reconstructed sample of each feature sample is $T_K$ and $f(t_k)$ denotes the feature value obtained at the sample, where $F = [f_1, f_2, \cdots, f_K]$, $K$ is the number of samples. The feature order is divided into a smooth trend term $f_T(k)$ and a random residual term $f_R(k)$ using an exponential weighted moving average.

After calculating the above index formula for all feature sets, the features with a correlation lower than 0.1 and significantly poor robustness performance are eliminated so that the average performance of the feature set is improved, and the feature engineering is completed.

### 2.2. Fuzzy Broad Learning System

Chen [20] et al. proposed a comprehensive learning system to improve the network by expanding the width of the learning network. The width structure of the network has a faster computational speed because there are no multi-layer connections. The FBLS [17] model eliminates the adjustment process of sparse auto-encoder in BLS, replaces the feature nodes of BLS with Takagi–Sugeno fuzzy subsystems, and then sends the intermediate outputs generated by all fuzzy subsystems as vector connections to the augmentation nodes for further nonlinear transformation. The fuzzy broad learning system improves the model's classification capability while considering the computational speed, and its basic structure and the fuzzy subsystem structure are shown in Figures 2 and 3.

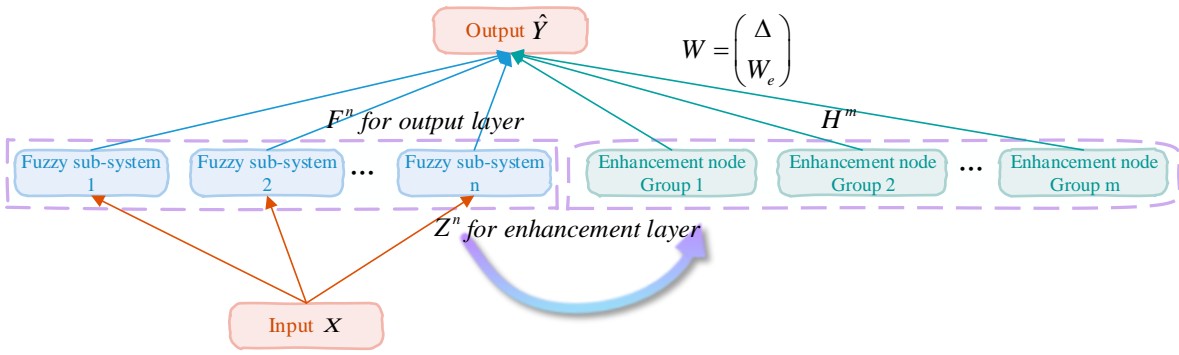

**Figure 2.** Fuzzy broad learning network structure.

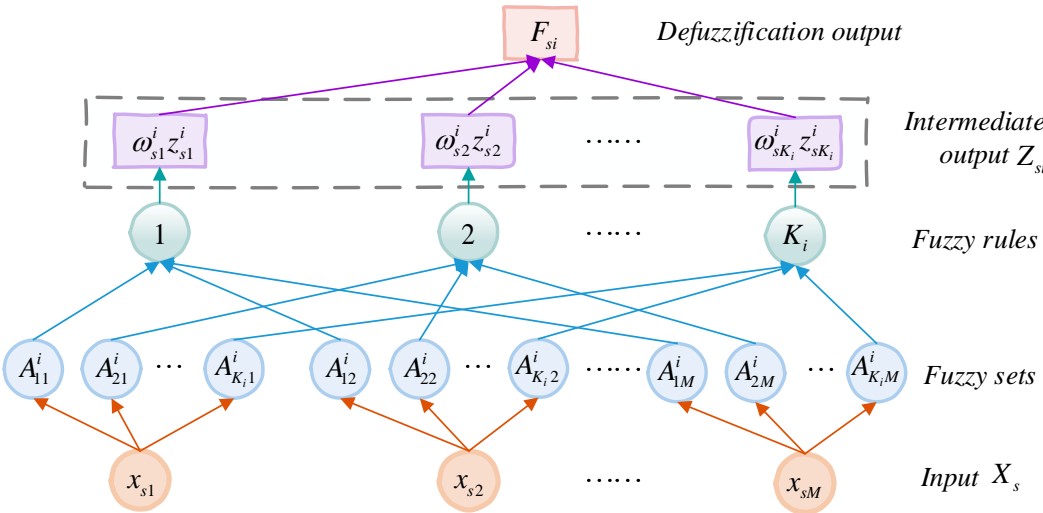

**Figure 3.** Fuzzy subsystem structure.

In Figure 2, it is assumed that there are $n$ fuzzy subsystems and $m$ groups of augmented nodes in FBLS. The input data are $X = (x_1, x_2, \cdots, x_N)^T \in R^{N \times M}$ and the sample features are $x_s = (x_{s1}, x_{s2}, \cdots, x_{sM})$, $s = 1, 2, \cdots, N$. In Figure 3, the $K_i$ fuzzy rule in the $i$th fuzzy subsystem is as follows: If $x_{s1}$ is $A_{k1}^i$, $x_{s2}$ is $A_{k2}^i$ and $x_{sM}$ is $A_{kM}^i$, then $z_{sk}^i = f_k^i(x_{s1}, x_{s2}, \cdots, x_{sM})$, $k = 1, 2, \cdots, K_i$, where $z_{sk}^i = \sum_{t=1}^{M} \alpha_{kt}^i x_{st}$, $\alpha_{kt}^i$ is the coefficient initialized by a uniform distribution of random numbers from $[0, 1]$ and then determined by the pseudo-inverse. In addition, we define the activation strength of the $k$th fuzzy rule in the $i$th fuzzy subsystem as $\tau_{sk}^i = \sum_{j=1}^{M} \mu_{kj}(x_{st})$, and then the weight of each fuzzy rule can be expressed as:

$$\omega_{sk}^i = \frac{\tau_{sk}^i}{\sum_{k=1}^{K_i} \tau_{sk}^i} \tag{6}$$

The chosen Gaussian affiliation function is defined as follows.

$$\mu_{kt}^i(x) = \exp\left(-\left(\frac{x - c_{kt}^i}{\sigma_{kt}^i}\right)^2\right) \tag{7}$$

where $c_{kt}^i$ is the center of the Gaussian affiliation function initialized by the k-means algorithm applied to the training set of the fuzzy subsystem and the clustering center. To simplify the model, $\sigma_{kt}^i$ is set to 1.

In order to retain more information implied by the input, the output vector of the $i$th fuzzy subsystem used for the training sample $X_s$ is denoted as $Z_{si} = (\omega_{s1}^i z_{s1}^i, \omega_{s2}^i z_{s2}^i, \cdots, \omega_{sk_i}^i z_{sk_i}^i)$. The output of the $i$th fuzzy subsystem of the training sample $X$ is $Z_i = (Z_{1i}, Z_{2i}, \cdots, Z_{Ni})^T \in R^{N \times K_i}$, $i = 1, 2, \cdots, n$. We define the intermediate output matrix of the $n$ fuzzy subsystems as $Z^n = (Z_1, Z_2, \cdots, Z_n) \in R^{N \times (K_1 + K_2 + \cdots + K_n)}$ and send $Z^n$ to the enhanced nodes for nonlinear transformation. Assuming that there are $L_j$ neurons in the $j$th augmentation node group, the enhanced node layer represents the output matrix as:

$$H^m = (H_1, H_2, \cdots, H_m) \in R^{N \times (L_1 + L_2 + \cdots + L_m)} \tag{8}$$

where $H_j$ is the output matrix of the $j$th augmented node group and $H_j = \xi\left(Z^n W_{hj} + \beta_{hj}\right) \in R^{N \times L_j}$, $W_{hj}$, and $\beta_{hj}$ are the randomly generated weights and biases. Consequently, the output of FBLS can be expressed as:

$$\hat{Y} = F^n + H^m W_e \tag{9}$$

where $W_e$ is the matrix of weights between the enhancement layer and the top layer, and the weights of the fuzzy subsystem to the top layer are set to 1.

## 3. Diagnostic Model

Bearings will affect the regular operation of the entire machinery if they fail when working with various mechanical components. Therefore, fault diagnosis of bearings needs to consider both diagnostic accuracy and diagnostic speed as much as possible. The bearing vibration signal has the characteristics of periodicity and instability, so it is necessary to select the main features of the original data that can better represent the signal through feature engineering and then apply these features to the prediction model to improve the prediction accuracy.

In addition, compared with deep learning methods, FBLS uses a combination of a fuzzy and broad learning system, which has a simple structure, high model training efficiency, and fewer network parameters. The FE-FBLS model proposed in this section is a solution to the problems of inaccurate feature extraction, a large number of hyperparameters optimization, and extended model training time in traditional bearing fault diagnosis methods, so it is feasible for bearing fault diagnosis. The specific workflow of the fuzzy broad learning system based on feature engineering for the bearing fault diagnosis method is shown in Figure 4.

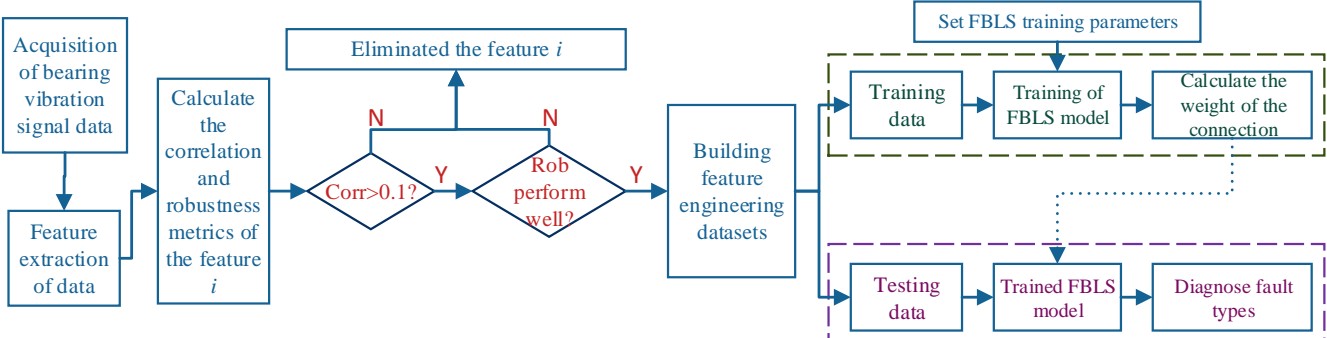

**Figure 4.** Fault diagnosis process based on feature engineering and FBLS method.

As illustrated in Figure 4, feature extraction of the bearing's vibration signal data is required first, which is an essential step before fault detection. After selecting the feature extraction method mentioned in Section 2.1 to extract the collected vibration signals, we calculated the correlation index value and robustness index value of each feature. Features that satisfy both screening conditions (that is, the correlation index value is greater than 0.1 and the robustness index remains at the ideal level) are chosen, and the combination of features that pass the screening is used to create a new feature set. The processed data are then divided into two groups: Training and testing. Training is performed on the original FBLS model using the training set data to determine the weights between the connected layers. Testing set data are then fed to the trained classification model to obtain the accuracy of fault classification and the time required to train the model.

## 4. Experimental

This experiment is divided into three parts, primarily for the faulty bearing data collected from the experimental bench, the centrifugal pump fault data from scholar Anil Kumar's experiment, and the public experimental data from Case Western Reserve

University to experiment on the proposed method. The parameter settings of the network model refer to the settings in the literature [17] and are adjusted on dataset 1 based on expert experience. The number of fuzzy rules for the fuzzy broad learning system is 80, and the number of enhanced nodes is 6; the number of feature nodes for the broad learning system is 500, and the number of enhanced nodes is 500. The model parameters used for each set of experiments were consistent. Considering that there have been many recent papers showing the advantages of deep learning in processing image data, the datasets in the deep learning models used in all the compared methods in this experiment are all processed image datasets.

The hardware and software environments in which the experiments were conducted are shown in Table 1.

**Table 1.** Experimental environment description.

|  | Part | Configured Version |
|---|---|---|
| hardware | CPU | Intel Core i9-10900 |
|  | GPU | NVIDIA Quadro RTX 4000 |
|  | Memory | 32 G |
|  | Operating System | windows10 64 bit |
| software | Python | 3.7.11 |
|  | Pytorch | 1.7.1 |
|  | CUDA | 10.2.89 |

### 4.1. DATASET 1

The vibration signal dataset collected by the QPZZ-II rotating machinery defect simulation experiment bench was used to generate the experimental dataset in this section. To detect the bearing vibration signal, the experiment employs a piezoelectric acceleration sensor type. The acquisition system is based on National Instruments' USB-4431 multifunctional high-precision data acquisition module. The acceleration sensor converts the signal to an electrical signal, then to a voltage signal via the charge amplifier, and finally to a digital signal via the data acquisition card. Finally, the code produced by Lab View software is attached to the data acquisition card for data gathering and storage.

The experimental bearing speed is 1188 r/min, the sampling frequency is 12 kHz, and the machine load is zero. The states of the experimental bearings include a 0.05 mm depth of roller failure, a 0.45 mm depth of roller failure, a 0.05 mm depth of inner ring failure, a 1.5 mm depth of inner ring failure, a 0.05 mm depth of outer ring failure, a 1.5 mm depth of outer ring failure, and bearings in the no-fault state. Figure 5 depicts the experimental platform and the defective bearing.

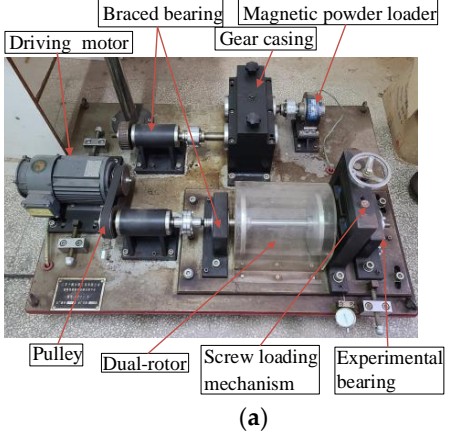

(a)

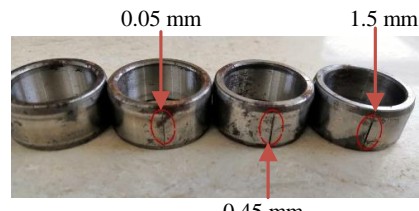

(b)

**Figure 5.** Experimental platform: (**a**) The test bench; (**b**) photographs of defective components under study.

Traditional deep learning often requires many samples to obtain favorable results. To illustrate, the FE-FBLS model is superior regarding the number of samples required. In this experiment, the number of samples per fault type input into the fuzzy broad learning system and the broad learning system is 100, and the total number of samples is 700. The number of samples per fault type input into the deep learning model is 300. The total number of samples is 2100, and the ratio of the training set and the test set is set as 8:2. The experiments were conducted six times, and the average value was taken as the final experimental results, which are shown in Table 2, Figures 6 and 7.

**Table 2.** Comparison of each classification model.

| Model | ResNet34-TL | MobileNet-TL | ResNet34 | MobileNet | CNN | FE-FBLS | FE-BLS |
|---|---|---|---|---|---|---|---|
| Accuracy of the test set | 96.84% | 93.02% | 87.82% | 73.54% | 90.03% | 96.43% | 96.19% |
| Training time/s | 52.2495 | 38.1495 | 50.8465 | 39.1945 | 51.123 | 0.2879 | 0.1054 |

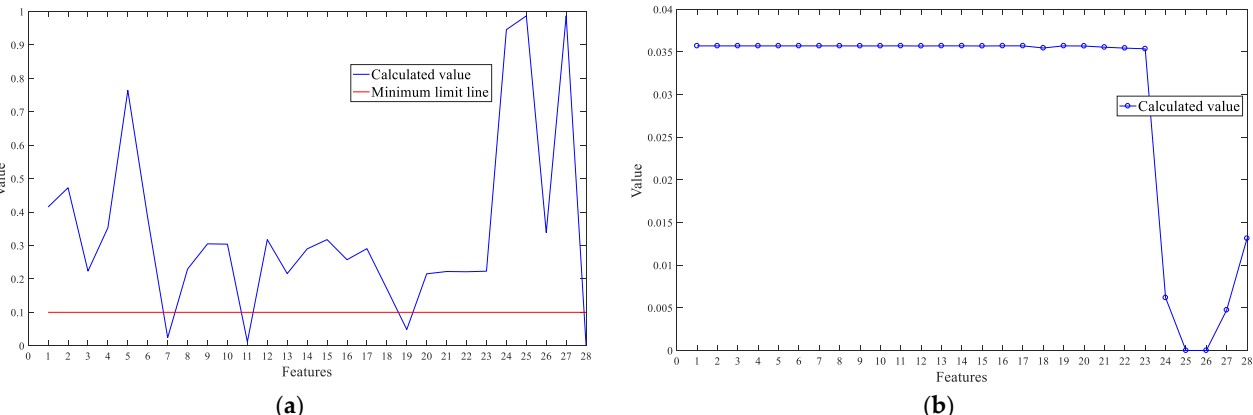

**Figure 6.** Value of each characteristic calculation index under Dataset 1: (**a**) Correlation; (**b**) robustness.

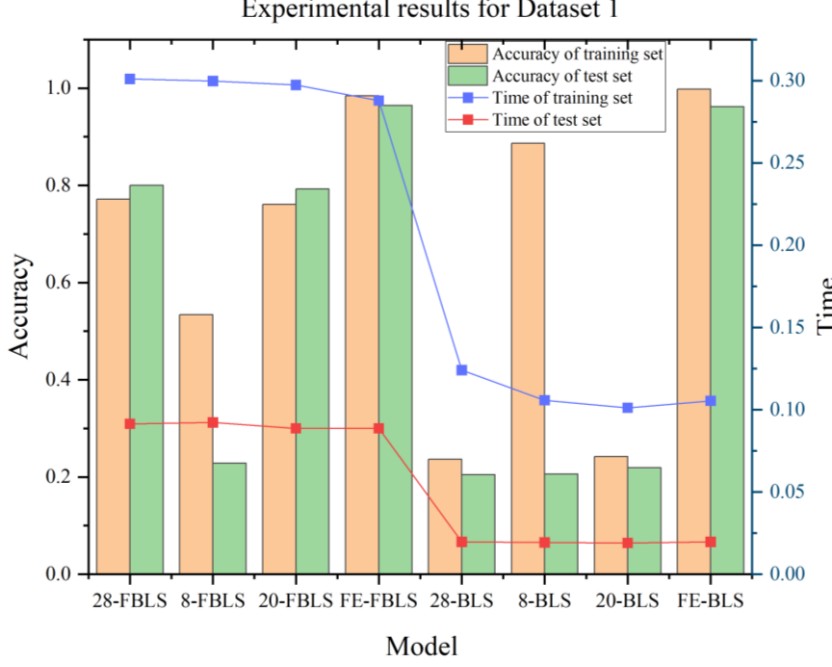

**Figure 7.** Experimental results of different features on dataset 1.

In the comparison experiments, the selected comparison methods are Convolutional Neural Networks [21], the deep Residual Network (ResNet) [22], and MobileNet [23] of the deep learning networks, and the last two deep learning models are optimized using parametric Transfer Learning (TL) [24] in order to optimize the selected networks. Then the selected deep learning methods for comparison are CNN, ResNet, MobileNet, the Residual Network based on Transfer Learning (ResNet-TL), and MobileNet networks based on Transfer Learning (MobileNet-TL), respectively. FE-FBLS and FE-BLS are the fuzzy broad learning system and the fuzzy broad learning system combined with feature engineering methods, respectively.

In the comparison method, the batch size of each deep learning model is set to 32, the number of iterations is set to 20, the learning rate is set to 0.0005, the dropout is set to 0.5, and the optimization method of the deep convolutional neural network is the Adam optimization algorithm.

The experimental results are shown in Table 2, and it follows from the results that the training time of the broad learning system and the fuzzy broad learning system is substantially reduced compared to the four deep learning models. The suggested FE-FBLS model can train the model in 0.29 s, but the deep learning network model takes a long time to train each generation. This is because the deep learning network model is developed from the network's depth, and the accuracy of the diagnostic model is improved by increasing the number of network layers to increase the depth of information obtained by the network, but the higher the number of network layers, the longer it takes to run. The fuzzy broad learning system is an enhancement of the Random Vector Functional-link neural network; therefore, it has a basic structure that allows for rapid training.

The transfer learning-based deep residual network has the highest diagnostic accuracy of each model at 96.84%, followed by the proposed FE-FBLS model at 96.43%. Fuzzy broad learning replaces each feature node with a separate Takagi–Sugeno fuzzy subsystem, and this improvement makes the fuzzy broad learning system more suitable for regression and classification applications, resulting in greater fault classification accuracy than the broad learning system. However, in the presence of massive data, the node selection and pseudo-inverse computation of FBLS randomness show that the accuracy of this network is lower than that of networks that use a depth structure. Based on the experimental results presented above, it is clear that the proposed model retains the advantage of being able to perform speedy diagnosis while ensuring model accuracy.

In addition, the second set of experiments is conducted in this section to verify the necessity of feature engineering methods for diagnostic models. The classification models using 28 hybrid features are named the "28-model", the classification models using 8-wavelet energy entropy features are named the "8-model", and the classification models using 20 time and frequency domain features are named the "20-model". The classification models using features obtained through feature engineering are named the "FE-model".

We calculated the correlation and robustness values of all the features, and the results are shown in Figure 6a,b. The horizontal coordinate of Figure 6a is the number of all features, and the vertical coordinate is the value of the correlation index corresponding to each feature in the range of [0, 1]. The horizontal coordinate of Figure 6b is the number of features and the vertical coordinate is the value of the robustness index corresponding to each feature in the range of [0, 0.04]. Then, we eliminate the features with correlation coefficients lower than 0.1 and those with significantly lower robustness than other features. The features numbered 7, 11, 19, 24, 25, 26, 27, and 28 are eliminated from this group of experimental data.

The results of the second set of experiments are shown in Figure 7. In terms of accuracy, it can be seen that the FE-FBLS model has the highest diagnostic rate on both the training and test sets, with 100% on the training set and 96.43% on the test set. This is because effective feature extraction can eliminate irrelevant data and information, compress effective information, and reduce the computational effort of information fusion, thus improving the real-time performance of information fusion. Using only a single time-domain

feature, frequency—domain feature, and wavelet packet decomposition feature cannot fully represent the fault feature information. However, if all features are mixed directly, it will lead to information redundancy and thus affect the diagnosis of the classification model. From the experimental results, it can be seen that a reasonable feature selection method can effectively improve the diagnostic accuracy of the fuzzy broad learning system.

*4.2. DATASET 2*

The experiment of this group uses the open dataset of rolling bearing fault diagnosis from Case Western Reserve University [25] to verify the stability of the proposed model. The sampling frequency is 12 kHz, the speed is 1797 r/min, the fault depth of each faulty bearing is 0.280 mm, and the fault width is 0.178 mm and 0.533 mm, respectively. Two fault sizes of the rolling element, two fault sizes of the inner ring, two fault sizes of the outer ring, and normal states are selected as the fault dataset, and the load environment is 0.

The number of samples per fault type input into the broad learning model is 50, and the total number of samples is 350. The number of samples per fault type input into the deep learning model is 300, and the total number of samples is 2100. The ratio of the training set and the test set is 8:2. The experiments were conducted six times, and the average value was taken as the final experimental results, and the results obtained from the experiments are shown in Table 3, Figures 8 and 9.

**Table 3.** Comparison of the accuracy of each classification model.

| Model | ResNet34-TL | MobileNet-TL | ResNet34 | MobileNet | CNN | FE-FBLS | FE-BLS |
|---|---|---|---|---|---|---|---|
| **Accuracy of the test set** | 99.66% | 98.44% | 97.18% | 92.8% | 97.56% | 100.00% | 100.00% |
| **Training time/s** | 52.744 | 37.6075 | 49.9525 | 37.8175 | 51.865 | 0.2624 | 0.0918 |

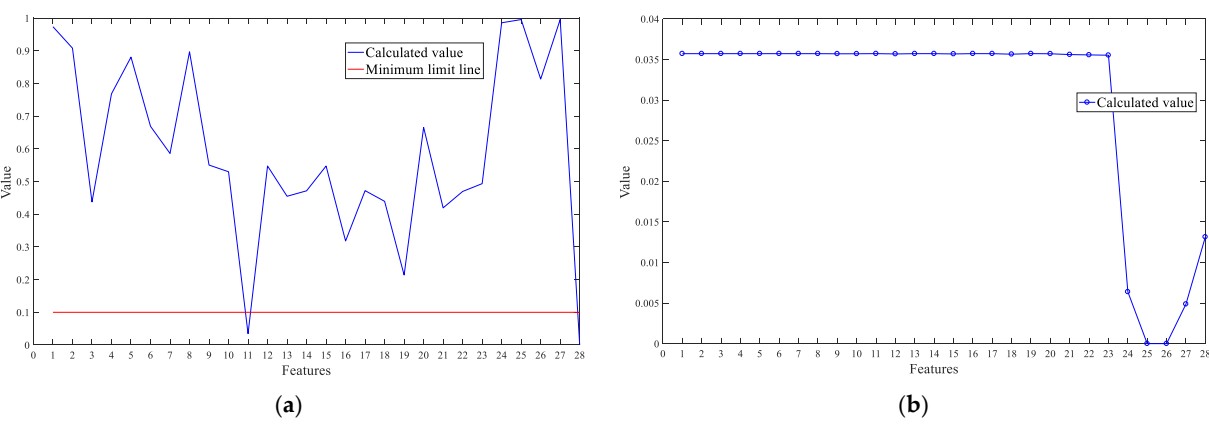

**Figure 8.** Value of each characteristic calculation index under Dataset 2: (**a**) Correlation; (**b**) robustness.

Table 3 shows an obvious advantage of the proposed diagnostic model in this set of experiments. With a reduced number of samples, the FE-FBLS model for the experiments using the publicly available dataset still identified the bearing fault types with an average accuracy of 100% and the training time of the model was only 0.26 s. It can be seen that the proposed model performs well on different datasets, which aptly reflects the feasibility of the fuzzy broad learning system as an alternative to deep learning networks. While the deep learning network model using transfer learning also improved the fault diagnosis accuracy using this experimental dataset, the accuracy rate fluctuates in each experiment and is not as stable as the fuzzy broad learning system, which is because the weight and bias matrix between the feature layer and the output layer of the fuzzy broad learning system is derived from the pseudo-inverse, so the results obtained when the model is classified again after completing the training will more stable. In addition, experiments

combining different features with the fuzzy broad learning system and the broad learning system were conducted again. The results are shown in Figures 8 and 9.

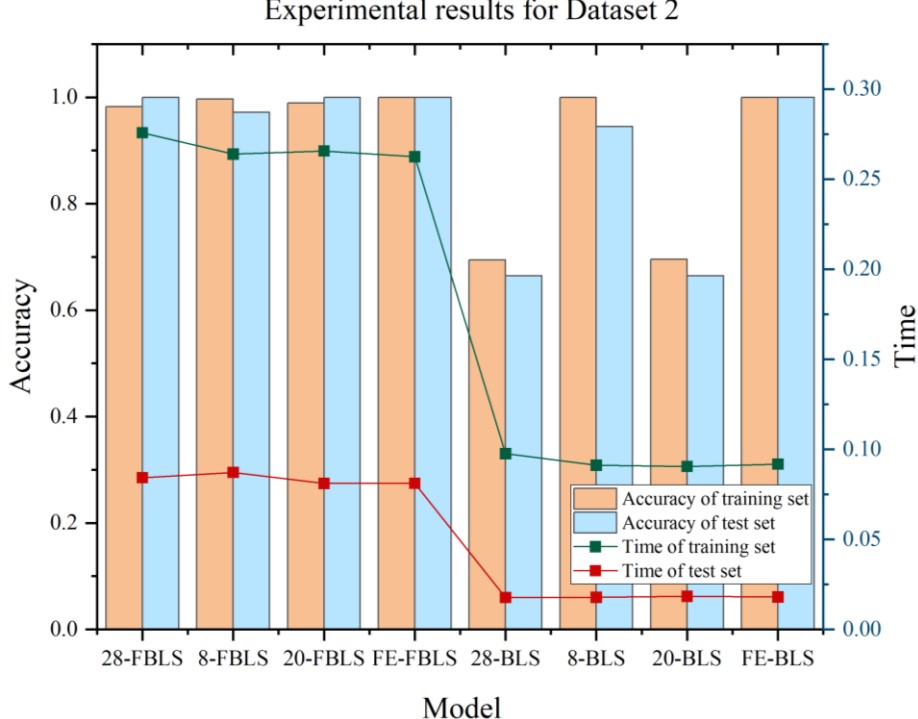

**Figure 9.** Experimental results of different features on dataset 2.

From the relevance and robustness values of the features evaluated by feature engineering represented in Figure 8, the meaning of each horizontal and vertical axis in Figure 8 follows the same meaning as the axes in Figure 6. It can be seen that the features to be eliminated are features 11, 24, 25, 26, 27, and 28. Then the remaining features are fed into the fuzzy generalized learning system.

From Figure 9, compared with other unscreened features, FE-FBLS has the highest fault diagnosis rate, which verifies the feasibility of the proposed feature extraction method. In terms of classification model selection, comparing the results obtained by combining FBLS and BLS with different feature sets reveals that the FBLS model with fuzzy rules has lower requirements on feature sets and better adaptability of the model, resulting in a higher fault diagnosis accuracy rate based on the FBLS model. However, the results of BLS combined with different features show that the necessity of features screened by the feature engineering described in this research is that too many or too few features will impair the model's classification performance to some extent. Only when the suitable feature set is chosen will the classification model with high feature requirements function well. The two sets of experimental results can reflect the rationality of the proposed method in the selection of features and classification models.

### 4.3. DATASET 3

In this section, vibration signals from the bearings of a centrifugal pump are selected for experiments to verify the applicability of the FE-FBLS model. Unlike the previous two sections, the data and comparison methods selected in this section are from the literature [26,27], and the centrifugal pump experimental bench is shown in Figure 10. A 0.37 kW electric motor drives the pump, the sampling frequency is 70 kHz, the pump's discharge is 1.6 L/s, and the diameter of the impeller is 119 mm. The speed of the pump was measured by a tachometer and was found to be 42 Hz. Photographs of a defective component under study are shown in Figure 11.

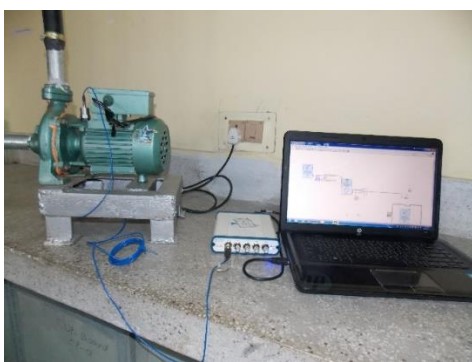

**Figure 10.** Centrifugal pump test bench.

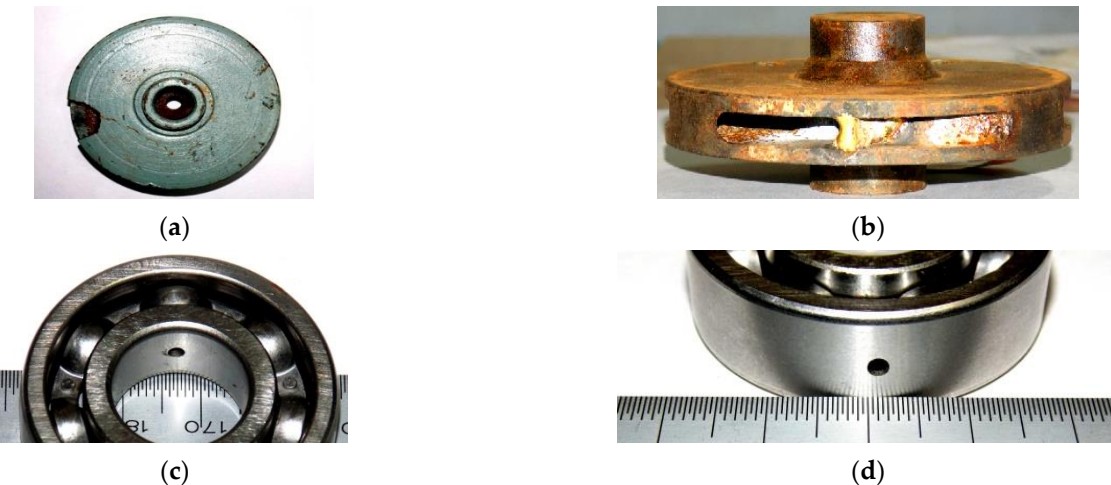

**Figure 11.** Photographs of the defective component under study: (**a**) Broken impeller (BI); (**b**) clogged impeller (CI); (**c**) bearing with inner race defect (IR); (**d**) bearing with outer race defect (OR).

The types of faults set by the test bench are defect-free (DF), broken impeller (BI), clogged impeller (CI), bearing with inner race defect (IR), and bearing with outer race defect (OR). In the literature [27], the vibration signal is transformed into an image signal by analytic wavelet transform, and 200 samples for each fault type, with a total of 1000 samples, are input into the deep learning model. In this section, the experiments are still conducted using one-dimensional vibration signals, with 100 samples for each fault type and 500 samples input into the broad learning model. The experiments were conducted six times, and the average value was taken as the final experimental result, and the results obtained from the experiments are shown in Table 4.

**Table 4.** Comparison of classification accuracy of each classification model.

| Model | Accuracy of the Test Set |
|---|---|
| ANN [28] | 86.4% |
| ANFIS [29] | 84.8% |
| SVM [30] | 93.0% |
| CNN | 96.8% |
| Improved CNN | 100% |
| FE-FBLS | 100% |
| FE-BLS | 100% |

As shown in Table 4, previous researchers [27] conducted experiments on the collected centrifugal pump bearing data using machine learning and deep learning methods. In

the experimental results, the deep learning algorithm showed superiority in diagnostic accuracy, but the method could not reach 100% accuracy until the CNN was improved, indicating that there is still much room for improvement in the traditional machine learning and deep learning methods. However, both the width learning system and the fuzzy width learning system can achieve 100% accuracy, and the set of data includes two kinds of fault data, broken impeller and clogged impeller of the centrifugal pump, in addition to bearing faults, which reflects the feasibility of the selected method in performing fault diagnosis of rotating machinery and also verifies the stability of the FE-FBLS model.

Then, the classification experiments were conducted again using the experimental data in this section, using different features. The results of the value of each characteristic calculation index under Dataset 3 and the experimental results under different features are shown in Figures 12 and 13, respectively.

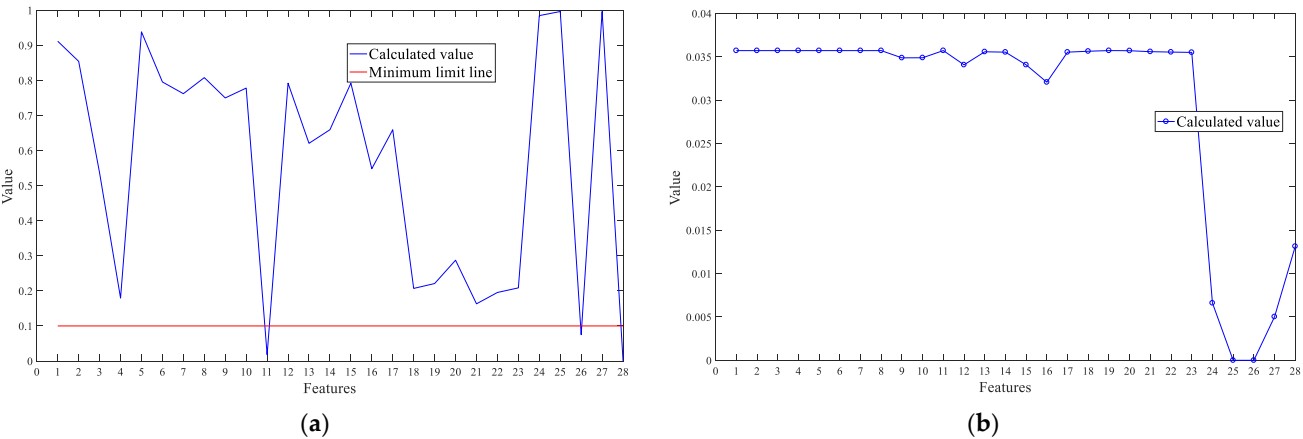

(**a**)　　　　　　　　　　　　　　　　　(**b**)

**Figure 12.** Value of each characteristic calculation index under Dataset 3: (**a**) Correlation; (**b**) robustness.

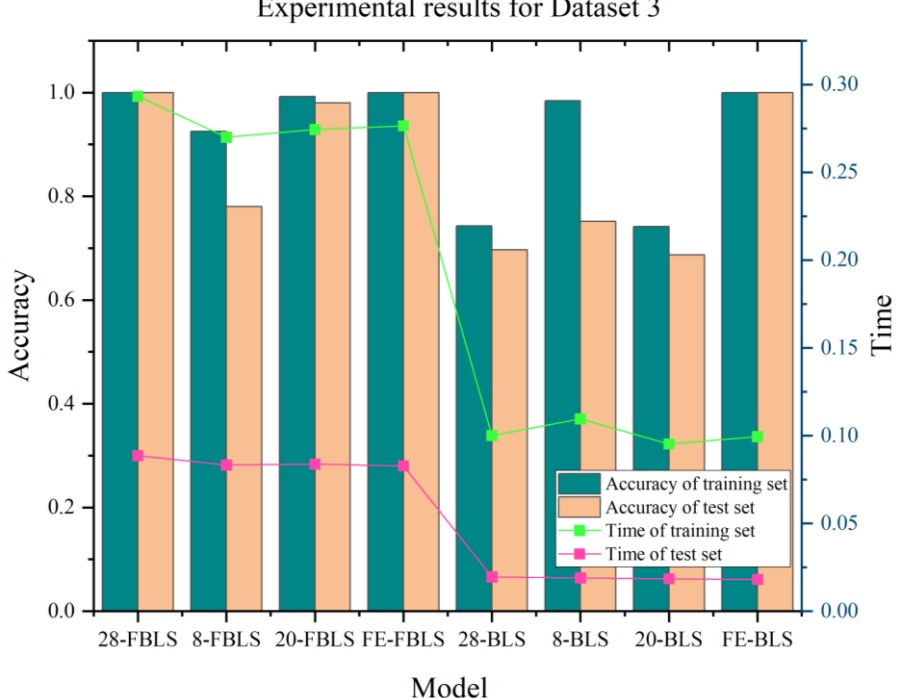

**Figure 13.** Experimental results of different features on dataset 3.

The meaning of each horizontal and vertical axis in Figure 12 has the same meaning as the axes in Figures 6 and 8. In Figure 12, features 11, 26, and 28 are below the restricted threshold of the correlation line, so these three features are eliminated; features 24 to

28 perform worse than other features of robustness index results, so they are also eliminated. Non-eliminated features formed a new feature set after feature engineering construction.

In Figure 13, the classification function of the broad learning system is influenced by the features of the bearing data of the centrifugal pump. The accuracy was significantly improved after using the feature engineering method. The fuzzy broad learning system performed well on different feature sets in Dataset 3 experiments, and the fault diagnosis accuracy reached 100% on both the training and test sets after being combined with the feature engineering method. In addition, the classification task of training and test samples could be completed within 1 s. This set of experiments further demonstrates the effectiveness of the proposed method for feature extraction and efficient fault diagnosis under different data and verifies that the fuzzy broad learning system can perform the classification task well when driven by bearing vibration data. Although in the above experiments, the fuzzy broad learning system uses fuzzy subsystems instead of mapping nodes and also eliminates the application of sparse self-encoders in the broad system, the required training time is longer than that of the broad learning system due to the change in the model structure. However, in practical applications, the proposed model already has the ability to perform the real-time diagnosis of bearings.

## 5. Conclusions

This paper proposes a method combining feature engineering and a fuzzy broad learning system for fault diagnosis of bearings, and its main contributions are.

1. Designed a feature engineering method of "inferiority" to make the extracted features more accurate and stable for bearing fault diagnosis by evaluating the indicators.

2. Replaced the feature nodes of BLS with Takagi–Sugeno fuzzy subsystems, used the FBLS system for bearing fault diagnosis, and verified the feasibility of the fuzzy broad learning model instead of the deep learning model

3. The effectiveness and robustness of the method were verified by three different sets of experimental data, which provided a new direction for researching bearing mechanical fault diagnosis. Compared with deep learning and machine learning, this model can perform fast and efficient fault diagnosis and is feasible in performing real-time bearing fault diagnosis for bearings.

This study provides a solution for intelligent fault diagnosis of rotating machinery and demonstrates the potential of fuzzy broad learning systems in fault diagnosis compared to other deep learning models. In future research, we will address the direction of model optimization by combining parameter optimization methods with fuzzy broad learning systems to reduce the dependence on manual labor.

**Author Contributions:** Conceptualization, X.Y. and J.Z.; methodology, X.Y.; software, X.Y.; validation, X.Y., L.L., and Y.W.; formal analysis, J.W.; investigation, G.H.; resources, Y.W.; data curation, L.L.; writing—original draft preparation, X.Y. and J.Z.; writing—review and editing, X.Y. and J.Z.; visualization, L.L.; supervision, J.Z.; project administration, X.Y.; funding acquisition, J.Z. All authors have read and agreed to the published version of the manuscript.

**Funding:** This research was funded by the National Natural Science Foundation of China, grant number 51865010; and the Science and Technology Project of Jiangxi Provincial Department of Education, grant number GJJ210639.

**Data Availability Statement:** Not applicable.

**Acknowledgments:** The work was supported by the National Natural Science Foundation of China.

**Conflicts of Interest:** The authors declare no conflict of interest.

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
