# Peer review of "Fuzzy Broad Learning System Combined with Feature-Engineering-Based Fault Diagnosis for Bearings"

_machines, doi:10.3390/machines10121229_

Round 1
Reviewer 1 Report
The paper proposes a method combining feature engineering and a fuzzy broad learning system for fault diagnosis of bearings, from the reading and review of the paper the following recommendations are made:
1. In lines 21-23 of the abstract it is recommended to expand the results obtained based on the advantages of the proposed method from numerical values and not simply from a statement, this will allow the reader to know what to find in the results and if it is attractive for its complete reading.
2. In figure 4 where the fault diagnosis process is illustrated, a clarification is required in the flowchart in the following sense: after calculating the correlation and robustness metrics of the feature i two verifications are made, in the upper part it is asked if corr>1 and in the lower part it is asked if Rob perform well, but it is not clear what the algorithm should perform from the flowchart in the case that one condition is satisfied and the other one is not satisfied.
3. The paragraph on lines 244-249 should be rewritten for better reading clarity as it is confusing wording.
4. At the end of lines 252 and 256 the words are split incorrectly, revise this aspect throughout the document to avoid this happening in other sections.
5. In figures 6, 8 and 12, the vertical axis should be clearly described in the type of units if possible and with a descriptive label.
Author Response
Thank you for your comments. The new manuscript has been revised. Looking forward to your reply.

Reviewer 2 Report
The submission can be published, as its technical merit is adequate to the level represented by an international journal, but after some improvements.
Some parts of the text require substantial extensions, e.g.:
-- the essential scheme of a diagnostic system, presented in fig. 4;
-- conclusions, summarizing each series of experiments (with three data sets) - in my opinion it is not enough to mention that the accuaracy reached 99-100%, but the Authors should answer the questions: why did we obtain such results, what is the reason of such a behaviour of experimental results, etc.?
The Authors should also improve some presentation drawbacks:
-- references are needed in the parts of the text which are taken from literature, e.g. description of wavelet decomposition (section '2.1. Feature engineering');
-- references are also needed e.g. in the paragraph in lines 237-243, as the names (e.g. ResNet34-TL) are not identified;
-- there are also other unexplained terms, e.g. MIV in line 53.
Author Response

(The authors gave the same response as above.)
